# Unsupervised identification of disease states from high-dimensional physiological and histopathological profiles

Kenichi Shimada[*] (iD) & Timothy J Mitchison

## Abstract

The liver and kidney in mammals play central roles in protecting the organism from xenobiotics and are at high risk of xenobiotic-induced injury. Xenobiotic-induced tissue injury has been extensively studied from both classical histopathological and biochemical perspectives. Here, we introduce a machine-learning approach to analyze toxicological response. Unsupervised characterization of physiological and histological changes in a large toxicogenomic dataset revealed nine discrete toxin-induced disease states, some of which correspond to known pathology, but others were novel. Analysis of dynamics revealed transitions between disease states at constant toxin exposure, mostly toward decreased pathology, implying induction of tolerance. Tolerance correlated with induction of known xenobiotic defense genes and decrease of novel ferroptosis sensitivity biomarkers, suggesting ferroptosis as a druggable driver of tissue pathophysiology. Lastly, mechanism of body weight decrease, a known primary marker for xenobiotic toxicity, was investigated. Combined analysis of food consumption, body weight, and molecular biomarkers indicated that organ injury promotes cachexia by whole-body signaling through Gdf15 and Igf1, suggesting strategies for therapeutic intervention that may be broadly relevant to human disease.

**Keywords** automated diagnosis; body weight loss; data mining; ferroptosis; toxicogenomics

**Subject Categories** Computational Biology; Molecular Biology of Disease; Systems Medicine

**Mol Syst Biol. (2019) 15: e8636**

## Introduction

A major function of the liver and kidneys is to take up, metabolize, and excrete xenobiotics that gain access to the blood. In executing these functions, these organs are at high risk for toxin-induced damage. Liver and kidney toxicities are major concerns in the safety of pharmaceuticals, industrial chemicals, and environmental toxins

(Stickel *et al*, 2000; Onakpoya *et al*, 2016). Predictive toxicology aims to identify molecular events that precede and cause tissue injury to inform exposure the limits and development of less-toxic alternatives (Suter *et al*, 2004; Chen *et al*, 2014). Toxicogenomics, i.e., collection of transcriptomic and other systematic data across sets of reference toxins in model organisms, is a relatively recent innovation with high potential for improving the mechanistic understanding of toxicities (Bredel & Jacoby, 2004; Chen *et al*, 2012). Here, we use the Open TG-GATEs database, which collates high-quality transcriptome, histopathology, blood chemistry, tissue, and body weight data following administration of 160 chemicals in rats (Igarashi *et al*, 2015; Data ref. Food consumption, 2015; Noriyuki *et al*, 2012). These data have previously been analyzed using gene-centric classification schemes with the goal of improving predictive toxicology (Natsoulis *et al*, 2008; Kohonen *et al*, 2017; Sutherland *et al*, 2018). Our approach is instead disease-centric, with more clinically oriented goals: to learn the number and nature of discrete disease states induced by toxins, how the liver and kidney respond to oppose induction of local pathology, and how they also orchestrate organism-wide responses to toxin exposure. We view this approach as conceptually similar to that of physicians seeking to classify the mechanisms of diseases more generally, and we ask, to what extent does unsupervised machine learning discover the same disease states as physicians?

Supervised classification, which dominates toxicogenomics, seeks to best separate multiple experimental outcomes (especially mRNA expression patterns) into pre-determined phenotypes (e.g., fibrosis or carcinogenicity in the liver; AbdulHameed *et al*, 2014; Eichner *et al*, 2014). It does not test whether the data best support those pre-determined phenotypes vs. others, or how multiple phenotypes relate to each other and interconvert (Kiyosawa *et al*, 2009; Sauer *et al*, 2017). We therefore sought first to identify mutually exclusive disease states in a data-driven manner and only then to decipher state-specific molecular mechanisms or biomarkers. To do this, we chose to begin with information a clinician can access in man, rather than classifying transcriptomes. Although transcriptomes offer large amounts of data and the potential for molecular pathway identification (Natsoulis *et al*, 2008; Sutherland *et al*, 2018), they are not part of standard clinical diagnosis, and their relationship to disease states is unclear. As a machine-mimic of clinical

Laboratory of Systems Pharmacology and Department of Systems Biology, Harvard Medical School, Boston, MA, USA
*Corresponding author. Tel: +1 617 432 6839; E-mail: kenichi_shimada@hms.harvard.edu

diagnosis, we started by clustering conditions that exhibit abnormal physiology (i.e., blood chemistry and body and tissue weights) and histopathology from the Open TG-GATEs dataset. These data collected in rats model a standard set of clinical measurements applied by physicians to patients with almost any unidentified disease. Using unsupervised clustering, we identified nine discrete disease states that were independently supported by physiological and histopathological data. We then performed a supervised analysis of gene expression data through the lens of these machine-identified disease states. Our combined data analysis revealed that some machine-identified disease states correspond to known disease states and known mechanisms of toxin action, but others seem novel. We identified temporal transitions between disease states that provide evidence for the induction of tolerance, and we found distinct gene expression signatures that correlated with tolerance. These included changes in the expression of xenobiotic metabolism genes, as expected, and also novel biomarkers for protection from ferroptosis, a specific form of cell death mediated by runaway lipid oxidation (Stockwell *et al*, 2017). Finally, we explored the role of the liver in mediating whole-organism responses to xenobiotics. We find evidence that the liver communicates with the rest of the body through specific signaling proteins that likely mediate feeding behavior and weight loss.

# Results

### Unsupervised identification of disease states from physiology and histology data

There is currently no standard method to classify toxin-induced pathology in a completely automated fashion. Therefore, we explored the physiology and histology space of an existing large toxicogenomic dataset (Open TG-GATEs), using unsupervised machine-learning techniques to identify disease states (Fig 1A; an overview of the data and our analysis workflow are summarized in Appendix Fig S1A and B; all the treatment conditions are listed in Dataset EV1). We performed an initial characterization of disease states using blood chemistry and body/tissue weight data (hereafter referred to as physiology data; Dataset EV2), which are unbiased measures corresponding to standard clinical tests used for diagnosis in patients. The dataset includes physiology data for 3,564 total conditions, representing administration of 160 chemicals at three dose levels each, with data collected at eight time points ranging from 3 h to 29 day (Fig EV1). To highlight dis-/similarity and partial dependencies between 37 physiology parameters across all 3,564 treatment conditions, we visualized these conditions using t-distributed stochastic neighbor embedding (t-SNE), which has recently gained popularity in biology, due to its success in dissecting heterogeneity of conditions from high-dimensional correlated data (van der Maaten, 2014). The t-SNE physiology map revealed one large island, which may correspond to normal physiology, and several small islands of abnormal physiology (Fig 1B, Appendix Fig S2). Naming of these islands, and their relationship to standard toxin-induced pathology states, is discussed below.

We next focused on liver and kidney histology data to ask whether they support the discrete disease states that emerged from physiology data (Dataset EV3). Histology is typically employed after

physiology to diagnose human disease. It is also part of the routine exercise required in regulatory toxicity assessments. In Open TG-GATEs, these data were recorded as calls from a standard constrained terminology made by expert toxicologic pathologists based on microscopic examinations of H&E-stained tissue sections. One compound treatment can induce more than one histopathology phenotype concomitantly, and typically, a tissue experiencing severe injury exhibits multiple histological phenotypes (Fig EV2, Appendix Fig S3). We computed "severity scores" based on the number of abnormal histology phenotypes scored. We then color-coded the physiology t-SNE map by liver and kidney histopathology severity scores (Fig 1C). Visual inspection confirmed good correlation between physiology and histology, where small distinct islands exhibited abnormal liver and kidney histology. However, conditions exhibiting similar physiological abnormality did not always agree on the histopathological severity scores. We suspected that this is in part due to lack of reliable severity metrics in histopathology; histopathology calls made by human experts are intrinsically less quantitative than physiology metrics (e.g., low-level changes may be missed) or may suffer from bias and/or inaccuracy due to scoring only a few tissue slices per animal. To overcome such potential missingness of histopathology, we imputed severity scores by smoothing the liver and kidney severity score distribution on the physiology map, where we can expect to see better agreement between the physiology and histopathology data (Fig 1D).

We next called disease states by identifying discrete clusters in the conjoined map of physiology and histopathology metrics using a density-based clustering algorithm (Ester *et al*, 1996). While t-SNE has been widely used due to its power to highlight the heterogeneity of the high-dimensional data in a lower dimensional space, its stochastic algorithm gives an output that are similar but slightly distinct from each other, depending on the pseudorandom number generator it uses. To compensate this stochastic nature of t-SNE, we ran t-SNE and clustering with different pseudorandom generator seeds iteratively for 100 times and sought for "consensus clusters" across the 100 different sets of clusters (Hornik, 2005). Through this procedure, nine consensus clusters emerged robustly, which we termed "disease state (DS)" clusters (Fig 1E). While use of the word "disease" implies a negative health impact, it is in principle possible that some "disease states" are beneficial. Of 3,564 conditions, each DS contained 37–203 conditions. 2,723 conditions did not belong to any DSs, and we classified these as non-disease state (non-DS; Dataset EV4). DSs are listed in Table 1.

### Known and novel disease states identified by unsupervised analysis

We next tried to map DSs onto standard pathologies in the toxicology literature, referring primarily to the physiology and histology markers, and in some cases also to subsequent analysis of gene expression changes or toxin mechanisms (described below). Comparison of histopathological severity scores showed which tissues were more affected in each DS (Fig 2A). A simplified clustergram highlighted physiology parameters that most strongly defined each state (Fig 2B). As we hoped, a distinct set of histology phenotypes were associated with DSs (Fig 2C).

Table 1 lists the nine DSs. While DS1–4 induced systemic physiological and pathological changes, they were not associated with

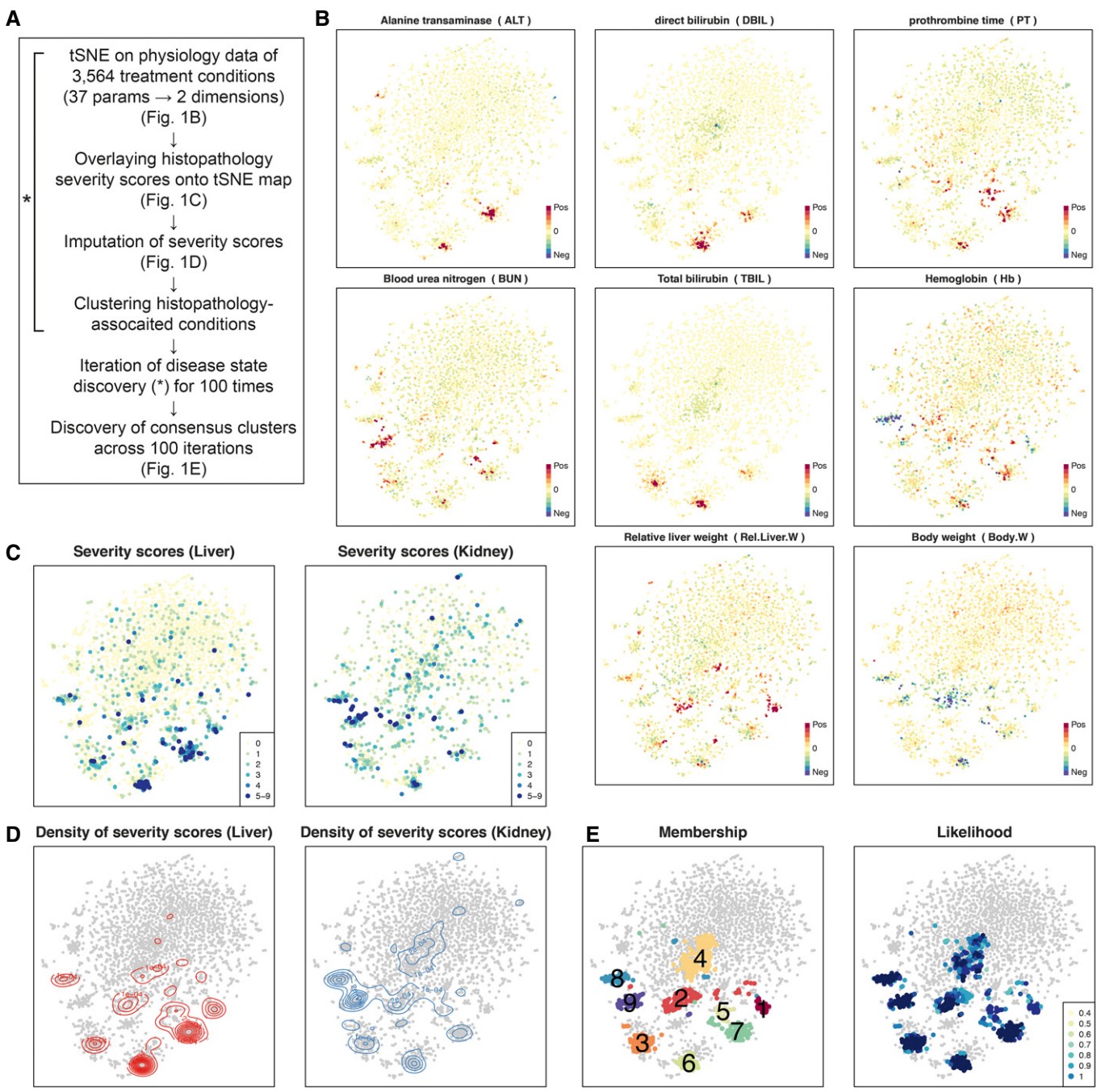

**Figure 1. Discovery of nine disease states using physiology and histopathology data.**

A   Computational process to discover disease states in Open TG-GATEs.

B   Physiology t-SNE map generated by Pearson distance and t-SNE. Each of 3,564 points represents the physiology data from one treatment condition. Plots were color-coded by intensities of eight physiology parameters.

C   Severity scores of liver and kidney overlaid onto the physiology tSNE map.

D   Contour maps indicating densities of severity scores in liver and kidney.

E   Membership and likelihood of consistent clusters across 100 times iterations of tSNE and DBSCAN clustering. Numbers on the clusters correspond to their names, DS1–9, that are consistent throughout the paper.

detectable tissue injury or exhibited transcriptional response to proinflammatory cytokines and would likely elude a conventional toxicogenomic analysis. DS5–9 showed clear signals of tissue injury

as scored by physiology and histology, and map clearly onto the kinds of disease used as pre-determined classifiers in conventional toxicogenomics. Recognition of DS1–4 therefore represents a

**Table 1. Characteristics of the nine disease states.**

| DS | Clinical description | Physiology | Histopathology | Induced transcriptome | Toxin class | Kinetics |
|----|---------------------|------------|----------------|----------------------|-------------|----------|
| 1 | Metabolically active | Rel.Liver.Weight up | Granular eosinophilic degeneration (L) | Anti-inflammatory; anabolism of ATP, lipids, amino acids (L) | Lipid-lowering drugs | Late-onset; robust |
| 2 | Tolerance | Body.Weight, GLC down | Hypertrophy (L) | Xenobiotic metabolism, ferroptosis resistance (L) | Synthetic hormones | Late-onset; robust |
| 3 | Hemolytic anemia | TBIL up | N/A | N/A | N/A | Early-onset; transient |
| 4 | N/A | TBIL, DBIL down | N/A | N/A | N/A | Early-onset; transient |
| 5 | Acute liver injury | PT up; TC, PL down | Eosinophilic change (L) | IL-1/TNFα signal, RNA-pol II transcription (L) | N/A | Early-onset; transient |
| 6 | Cholestasis | DBIL up | Proliferation of bile ducts (L) | IL-1/TNFα signal, cancer signature, response to alkaloid (L) | N/A | Early-onset; transient |
| 7 | Hepatocellular injury | AST, ALT, LDH up | Single cell necrosis (L) | IL-1/TNFα signal, cancer signature, cell cycle, collagen biosynthesis (L) | N/A | Early-onset; transient |
| 8 | Bleeding | WBC, Ret, Neu up; Hb, Ht, Lym, RBC down | Extramedullary hematopoiesis (L) | IL-1/TNFα signal, IL-6/Jak/STAT signal, complement/coagulation cascade (L) | NSAIDs | Late-onset; robust |
| 9 | Kidney injury | BUN, Rel.Kidney.Weight, up | Hypertrophy (K) | IL-1/TNFα signal (K) | N/A | Late-onset; robust |

This table summarizes the analysis of physiology (Fig 2B), histology (Fig 2C), transcriptome (Fig 3), toxin class (Fig EV3), and kinetics (Fig 4). Physiology acronyms are described in Dataset EV2. Refer to the corresponding figures and the text for the details.

preliminary success of our analysis, and DS2 in particular emerged as mechanistically significant in subsequent analysis. DS1 showed activated synthesis of various metabolites and increased relative liver mass. DS2 represents acquired toxin-induced tolerance while decreasing body weight. In DS3, total bilirubin (TBIL) increased, but direct bilirubin (DBIL), a liver injury marker, did not. Based on this profile, we suspect that DS3 corresponds to hemolytic anemia. In DS4, TBIL level was decreased, which is not seen in known clinical states. DS4 lacked common histological or transcriptome changes in liver and kidney and may not be a disease state in the conventional sense, though it clearly is abnormal (Hirayama *et al*, 1974) and reproducible. DS5–7 was marked by an increase in standard liver injury markers that are used in human diagnosis of liver damage (Giannini *et al*, 2005). DS5 exhibited longer prothrombin time (PT), suggesting a decrease in the synthesis of prothrombin, a liver-synthesized blood coagulation protein. DS5 would not clinically be considered to be an explicit liver injury, unlike DS6–7, but the data strongly suggest that the liver's health is affected. DS6 exhibited DBIL increase and various periportal histopathology phenotypes, which corresponds to cholestasis, an injury of the liver bile ducts. DS7, on the other hand, exhibited hepatocellular injury such as increase in blood asparatate aminotransferase (AST), alanine aminotransferase (ALT), and lactate dehydrogenase (LDH) levels as well as single cell necrosis in the liver. In DS8, multiple hematological parameters were changed and unique histological and transcriptional phenotypes were observed, indicating that the animals suffered from bleeding, induction of synthesis of complement factors and coagulation cascade components, and hematopoiesis in the liver (Gwaltney-Brant, 2014). DS9 is marked by blood urea nitrogen (BUN) increase and hypertrophy and neutrophil infiltration

in kidney, indicating kidney injury. The database contained fewer reference kidney toxins than liver toxins, perhaps explaining why we observed only one disease state that mapped to kidney pathology (Schrier *et al*, 2004) in the analysis.

### Overrepresentation of drug classes in disease states

We next asked whether specific toxin classes reliably induced specific DSs, defining classes as containing multiple compounds with known overlapping biological activity, as summarized in Appendix Fig S1C. We found that lipid-lowering drugs mapped to DS1, synthetic hormones mapped to DS2, and non-steroidal anti-inflammatory drugs (NSAIDs) mapped on DS8, each at higher doses, and each more than expected by chance (Fig EV3, Table 1). NSAIDs are known to cause intestinal bleeding at high doses (Wallace *et al*, 2000; Wallace, 2008), which likely accounts for their mapping to DS8 (Appendix Fig S4A and B). The four lipid-lowering drugs mapped onto DS1 were peroxisomal proliferator-activated receptor alpha (PPARα) agonists (clofibrate, fenofibrate, WY-14643) and a cholesterol synthesis inhibitor (simvastatin). PPARα agonists increased peroxisomes, which were recognized as eosinophilic granules in the cells in DS1 (Ohta *et al*, 2009; Appendix Fig S4C and D). This class of drugs has been well studied in rats, and its lipid-lowering effect in the short term is often described as beneficial rather than pathological. However, we decided to keep our initial notation of "disease states" even for DS1; long-term treatment with these drugs frequently causes liver cancers, possibly due to hyperactivation of metabolism to increase the liver biomass (Holden & Tugwood, 1999). The NSAID- and fibrate-induced disease states of liver were previously characterized by transcriptome-centric analysis

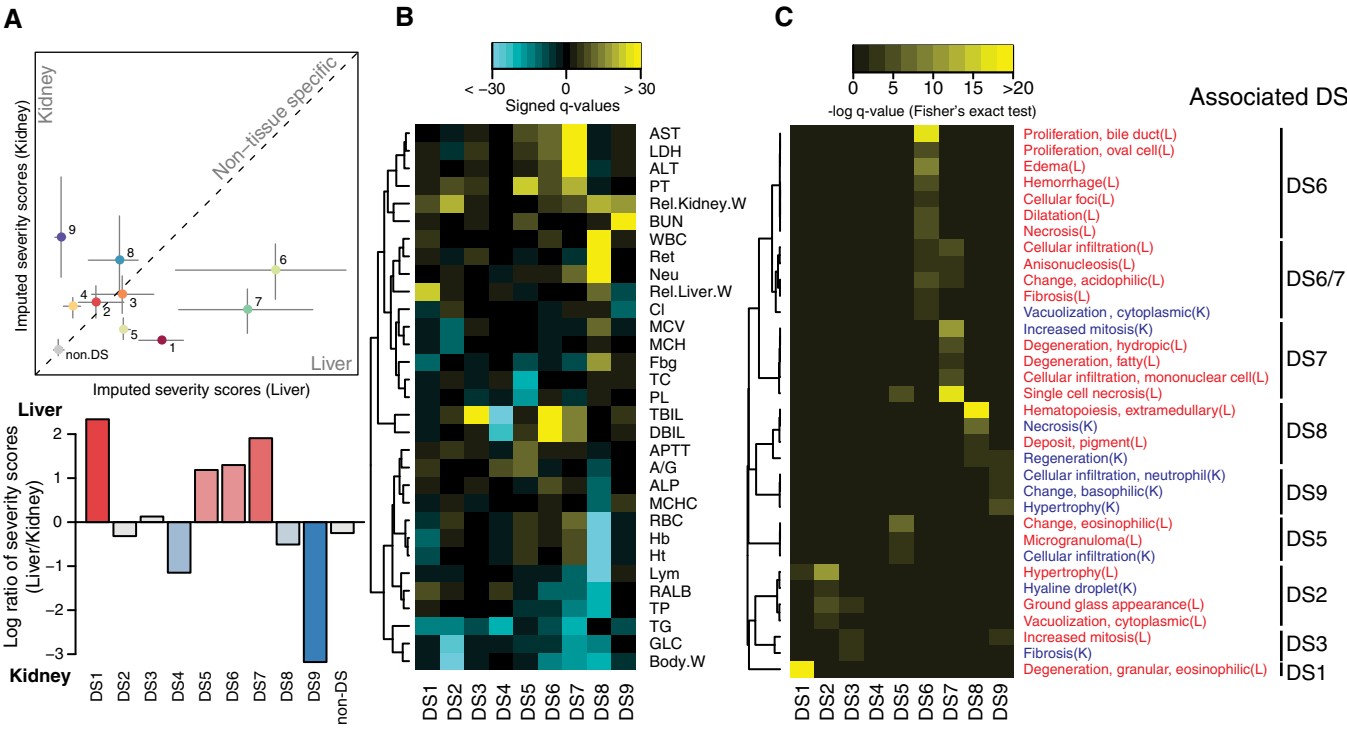

**Figure 2. Disease state characterization using blood physiology and histopathology.**

A  Comparison of liver and kidney severity scores (top). Imputed severity scores shown in Fig 1D were grouped for each DS, and their median (point) and lower and upper quartiles (bar) are shown. *X*- and *y*-axes correspond to liver and kidney severity scores, respectively. The number of samples per DS vary between 37 and 198. Non-DS has 2,723 samples. Log ratio of liver over kidney severity scores (bottom).

B  Changes in physiology parameters and DSs. Yellow/cyan indicates the parameters are higher/lower than control vehicle treatments in the DS. Thirty-one parameters whose FDR-adjusted *P*-values $< 1 \times 10^{-10}$ (Wilcoxon two-sample test) in $\geq$ one DS were shown.

C  Histopathology phenotypes and DSs. Relative enrichment of a phenotype among DSs was shown. Yellow indicates more observations in a DS than the others. Thirty-four histopathology phenotypes whose FDR-adjusted *P*-values $< 5 \times 10^{-3}$ (Fisher's exact test) in $\geq$ one DS were shown.

---

of Open TG-GATEs data (Chung *et al*, 2015). Compared to the two drug classes that strongly induced representative physiology and histology phenotypes of the DSs, the effect of synthetic hormones on liver was modest and lacked conspicuous physiology or histology changes such as hypertrophy (Appendix Fig S4E and F).

**Transcriptome description of disease states**

To test whether each DS was transcriptionally distinct, we performed elastic net classification of liver and kidney transcriptome data (Datasets EV5 and EV6). This generated classifiers that attempted to distinguish conditions assigned to each DS from all the rest of conditions using the liver or kidney transcriptome. All classifiers were found to be powerful for separating DSs (all areas under ROC curves were above 0.85, compared to 0.46–0.63 for randomly drawn samples of the same sizes; Appendix Fig S5A). Thus, each DS has a characteristic transcriptome both in liver and in kidney. To examine the functional implications of DS-specific transcriptional states, we assessed whether 914 GO and KEGG pathways changed their activities in DSs, compared to pooled non-DS clusters. We computed "activity scores" by modifying the standard gene set enrichment analysis (GSEA) to capture the significance of systematic changes of pathway activity among conditions assigned to each DS (See Materials and Methods for the detail). We interpreted a large positive or negative activity to

indicate that a pathway is substantially up- or downregulated compared to corresponding vehicle treatments in the DS. Through this analysis, we found that six DSs (DS1,2,5–8) induced systematic changes of pathway activities in the liver transcriptome; in DS9, changes of pathway activities were observed in the common kidney transcriptome but not in the liver transcriptome; and DS3–4 did not capture any common pathway changes either in liver or in kidney (Fig 3A). Hierarchical clustering of pathway activity in both liver and kidney primarily divided nine DSs into two groups (tissue injury or not; Fig 3B); consistently, five DSs associated with tissue injury activated proinflammatory cytokine signaling, namely cellular response to proinflammatory cytokines (TNF, IFN-γ, IL-1) in the corresponding tissues (Appendix Fig S5B).

Liver was more transcriptionally responsive to xenobiotic stimuli than kidney across many conditions that induced a DS. This is expected because the liver is primarily responsible for detoxifying xenobiotics, although the choice of toxins in the dataset may also overemphasize the role of the liver. The kidney only exhibited major transcriptional changes in DS9, which also phenotypically corresponded to kidney damage in the physiology and histopathology (Table 1).

To better understand the changes induced by toxin exposure in the six DSs we identified as significantly affecting the liver transcriptome (DS1,2,5–8), we classified the modulated pathways based

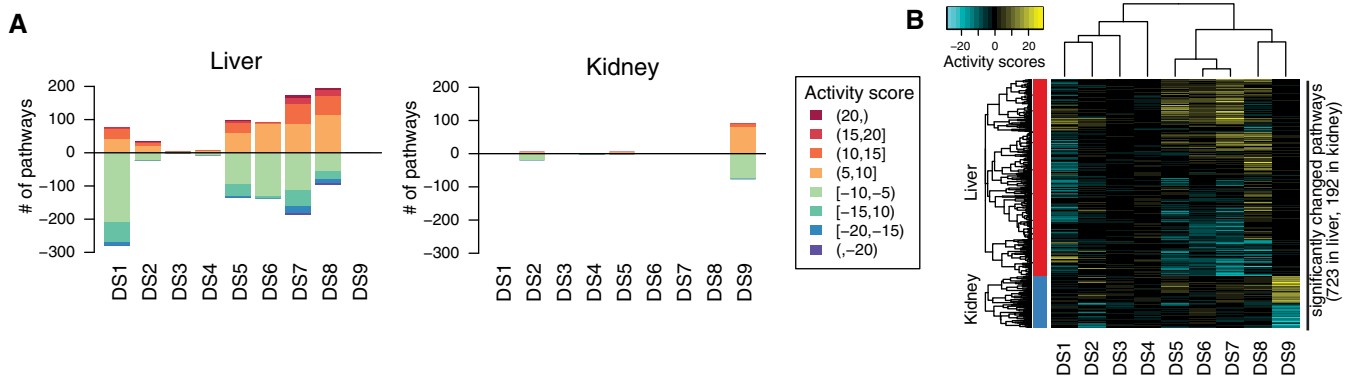

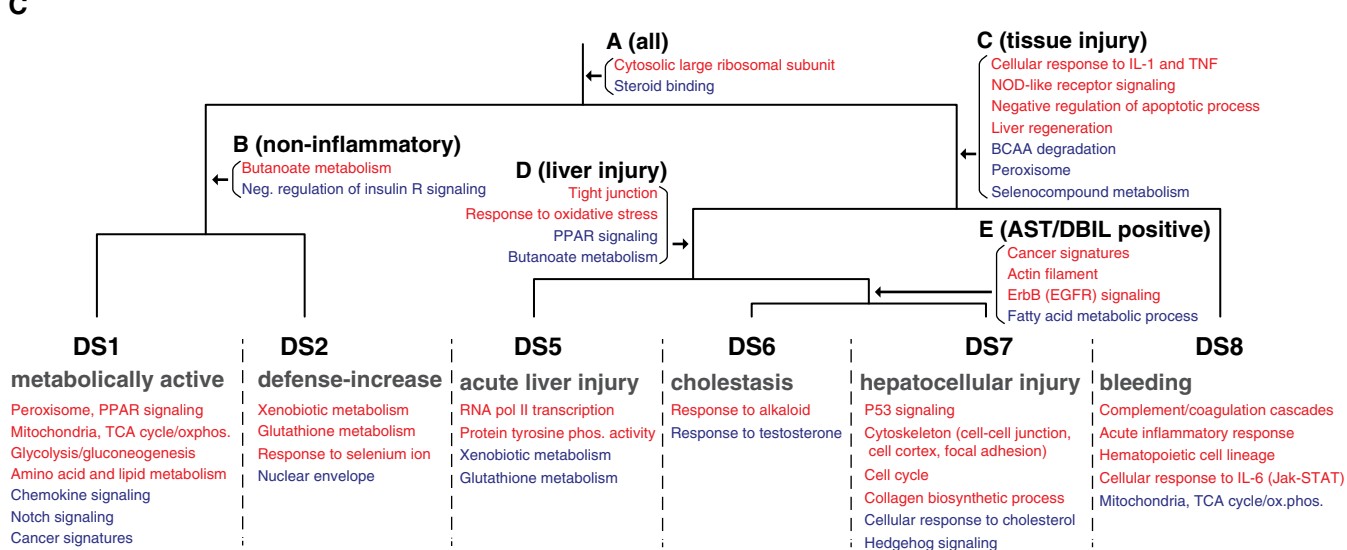

**Figure 3. Disease state characterization using transcriptome data.**

A  Number of significantly up-/downregulated pathways in each disease state from the liver and kidney transcriptome. Colors of the bars indicate activity scores of pathways.

B  Activity scores of significantly up-/downregulated pathways in the liver and kidney across all nine DSs.

C  Manually curated map of transcriptional activities of pathways in the liver among six DSs (DS1–2,5–8). Up- and downregulated pathways are colored in red and blue, respectively. See also Appendix Fig S5.

on their transcriptional activity. Pathways that were transcriptionally activated or suppressed in each DS were mapped onto corresponding nodes on a dendrogram (Fig 3C, Appendix Fig S5C–E; Dataset EV7). This allowed us to determine in which contexts the pathways were up- or downregulated. For example, in all of these six DSs, the liver activated transcription of "cytosolic large ribosomal subunit"; in disease states that reported on tissue injuries (DS5–8) pathways categorized as "cellular response to IL-1 and TNF" was activated in the liver. On the other hand, some pathways such as "p53 signaling" or "collagen biosynthetic process" were activated only in specific states [e.g., hepatocellular injury (DS7)], but not in the other DSs.

## DS2 is a state of drug-induced tolerance

A strength of the Open TG-GATEs data is the collection of multiple time points at constant toxin exposure. Kinetic analysis

showed that some DSs are mostly or exclusively late-onset, i.e., after 24 h (DS1,2,8). Others are early-onset or had no particular trends in kinetics (Fig 4A, Appendix Fig S6A). Of 365 conditions (compound and dose) whose data were collected at all eight time points, 246 (67%) caused at least one DS at some point, and 90 (25%) caused more than one DS over time (Appendix Fig S6B).

To inform on causal connections between DSs, we focused on conditions causing more than one DS and classified temporal transitions at constant toxin exposure (Fig 4B). This analysis revealed bi-directional inter-conversion between different DSs corresponding to liver damage, which may be expected given transcriptional and histologic overlaps between these related DSs. A strong, unexpected feature was conversion of multiple liver and kidney DSs to DS2. Since DS2 is not associated with liver or kidney injury by histology, this suggests a time-dependent reduction of organ-specific pathology. Figure 4C shows temporal

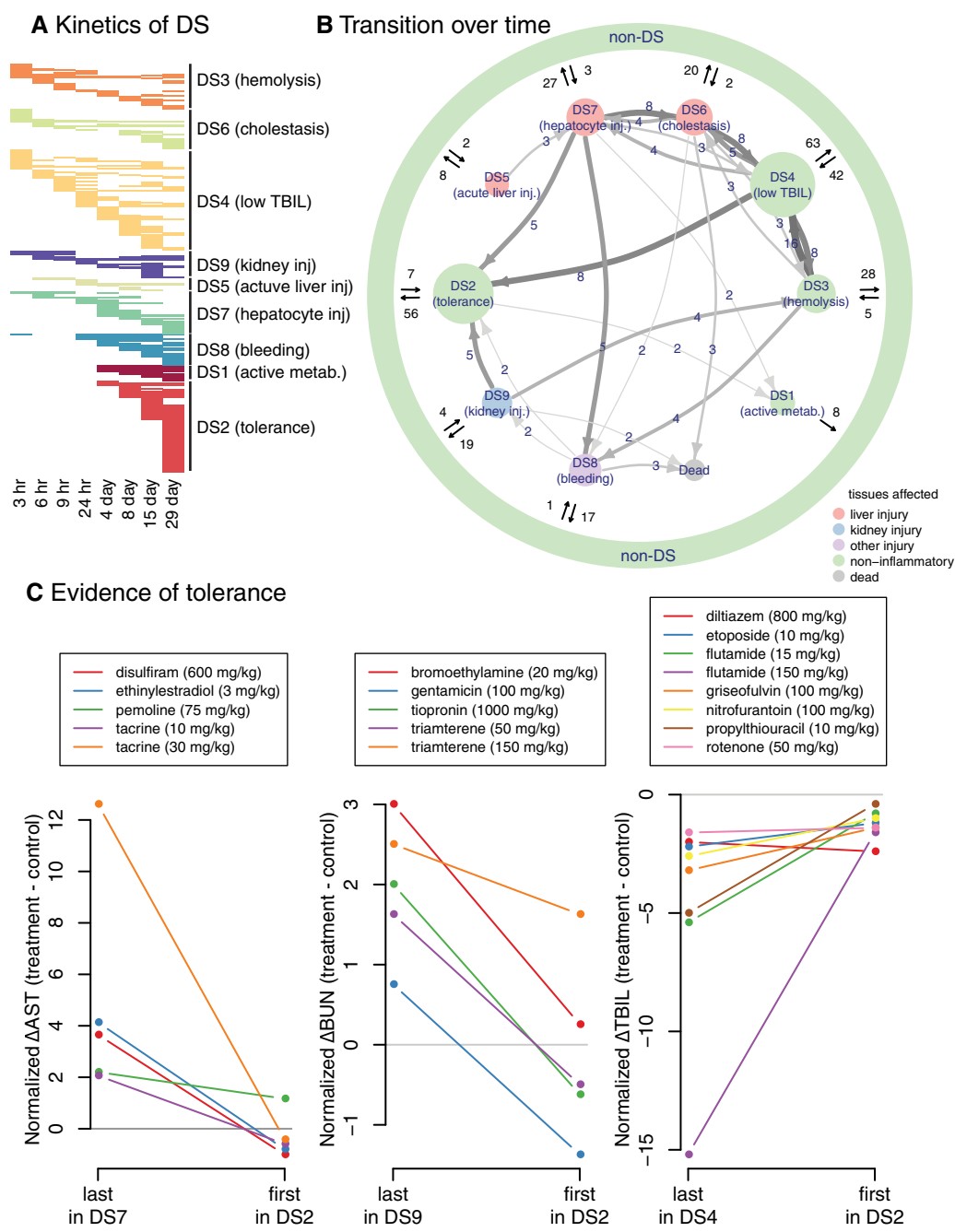

**Figure 4. Disease state dynamics.**

A  Kinetics of DS, summarizing the frequency and timing of each DS. Each row indicating one treatment (compound and dose) across eight time points.
B  Transition between DSs. Size of nodes reflects the number of unique treatments assigned to each DS. Size and color of arrows reflect the number of treatments transitioning between the two nodes. Transitions taken by only one compound were omitted.
C  Evidence of tolerance. As for transitions from three DSs to DS2, changes in representative markers for each DS were shown.

changes in a standard liver injury biomarker (AST), a standard kidney injury biomarker (BUN) and a representative biomarker of DS4 (TBIL) across the transitions out of other DSs and into DS2. Transitions into DS2 were accompanied by reduced injury biomarkers in most cases. After examining the transcriptome data on the different DSs (see below), we interpret this as evidence of induced tolerance.

Toxin-induced tolerance to toxin action, also known as autoprotection (Dalhoff Kim *et al*, 2003; O'Connor *et al*, 2014), is poorly understood but is presumably an important component of how xenobiotic resistance evolved, and how modern vertebrates adapt to toxic environments. Analysis of dynamics of DSs and injury biomarkers (Fig 4B and C, Appendix Fig S6C) suggests that DS2 is a state of induced tolerance. To determine molecular mechanisms that

might drive tolerance and result in DS2, we first re-examined genes that are selectively regulated in this DS. Xenobiotic catabolism genes were strongly and selectively induced in DS2 compared to all other DS and non-DS conditions (Fig 3C). While this finding might be expected, it emphasizes the function of xenobiotic defense genes and supports our characterization of DS2 as a state of tolerance.

### Tolerance is associated with induced resistance to ferroptosis

Xenobiotic metabolism is a multistep reaction: in phase I, cytochrome P450 monooxygenases (Cyp450) conjugate xenobiotic compounds with oxygen using NADPH; in phase II, the products of this reaction are conjugated with hydrophilic groups such as sugars (e.g., glucuronic acid) and glutathione (GSH) to facilitate excretion. When we regrouped xenobiotic catabolism genes based on cofactors, we found that genes encoding NADPH- and GSH-utilizing enzymes were among the most highly expressed genes in DS2 (Figs 5A and EV4A, Dataset EV8). Activation of redox metabolism functions is also involved in ferroptosis, a form of cell death that has been implicated in drug-induced liver injury (Lőrincz *et al*, 2015). Thus, we also suspected that the liver acquires resistance to ferroptosis in DS2.

Ferroptosis occurs when activity of the selenoprotein glutathione peroxidase 4 (Gpx4) is inhibited (Fig 5B; Yang *et al*, 2014). Ferroptosis pathways are not represented in current GO terms, in part because genes that regulate ferroptosis are still poorly understood (Stockwell *et al*, 2017). To generate an objective classifier of ferroptosis sensitivity, we referred to our previous pharmacogenomic analysis of the NCI-60 project, where we contrasted cellular responses to chemicals known to induce cell death via a number of mechanisms, including ferroptosis (Fer), DNA damage (DNA) and tyrosine kinase inhibition (TKI) in the NCI-60 human cancer cell-line panel (Shimada *et al*, 2016). Using those data, we generated gene expression signatures that serve as markers for sensitivity (sen) or resistance (res) for each of these treatments. We converted these six signatures to rat orthologs and evaluated them across all nine DS compared to non-disease samples. Fer-res was strongly and uniquely upregulated by DS2 (Fig 5C). Conversely, DSs associated with liver injuries downregulated Fer-res and upregulated Fer-sen, consistent with the role of ferroptosis in liver injury. DS2 does not increase DNA-res or TKI-res, suggesting that toxin-induced tolerance is associated with acquiring resistance to ferroptosis, but not to other cell death mechanisms.

There was no substantial overlap between genes involved in Fer-res and the 14 pathways activated exclusively in DS2 (Fig EV4B). Nevertheless, the transcriptional activity of Fer-res among 3,528 liver transcriptome is most highly correlated with the scores of the 10 GO and KEGG pathways exclusively upregulated in DS2 while the activity of Fer-sen is somewhat anticorrelated with them (Fig EV4C and D). Our analysis supports the hypothesis that DS2 not only overexpresses xenobiotic metabolism genes but also acquires resistance to ferroptosis.

### Whole-body response to toxins

Toxin responses have whole-body impacts in addition to organ-specific effects, but these have been much less studied in the toxicology literature, or the human liver/kidney disease literature. Loss

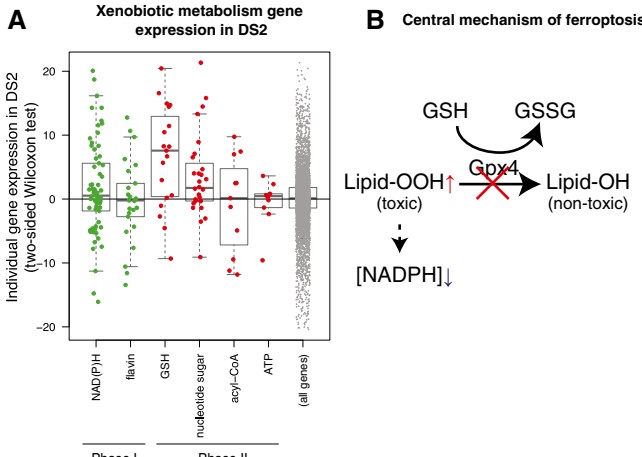

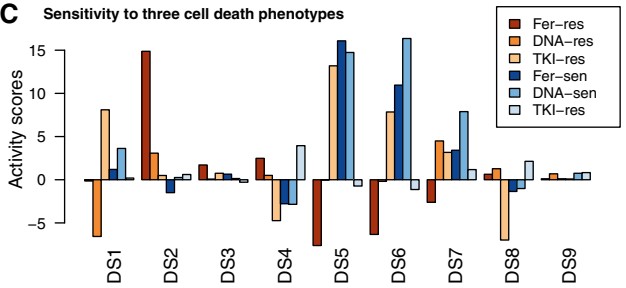

**Figure 5. Induced drug tolerance is partly due to resistance to ferroptosis.**

A  Expression of xenobiotic metabolism enzyme genes in the liver, grouped by phases of biotransformation and cofactors. Only cofactors with ≥ 10 enzymes were shown. *Y*-axis is a signed log *P*-value computed with two-sample Wilcoxon test between 126 conditions in DS2 and their corresponding vehicle treatment. The horizontal lines, box limits and whiskers represent the median, lower and upper quartiles and the most extreme data point which is no more than 1.5 times the interquartile range from the box.

B  Central mechanism of ferroptosis. Gpx4 reduces cytotoxic lipid peroxide to nontoxic alcohol species. When Gpx4 does not work, lipid peroxide is accumulated, and NAD(P)H is depleted consequently. Decrease in NAD(P)H or the cofactor of GPX4, GSH, makes cells susceptible to ferroptosis.

C  Activity of cell death mechanism-specific sensitive and resistant gene signatures in the nine DSs, focusing on three cell death mechanisms (Fer: ferroptosis, DNA: DNA damaging, TKI: tyrosine kinase inhibition).

of body weight is a well-known indicator of chronic toxicity in rats, though its etiology is poorly understood. It likely corresponds to cachexia phenotypes in man, which are well-recognized contributors to disease burden (Delano & Moldawer, 2006). Body weight decrease was the most significant physiological descriptor for DS2, but it was not unique to DS2: all DSs associated with tissue injury (DS5–9) exhibited body weight decrease to a similar extent to DS2 (Figs 6A and EV5A), while tolerance-associated pathways were activated in DS2, but not in DS5-9, highlighting the differences between the states (Fig 6B). Toxin-induced weight loss could result from direct responses of peripheral tissues such as muscle and adipose tissues to toxin, or (more likely) from their indirect responses to signals from toxin-exposed tissues that alter global metabolism or/and feeding behavior. The liver and kidney communicate with the

whole body through concentrations of multiple metabolites in blood and also through secreted signaling proteins. We therefore analyzed metabolite and secretome changes in conditions that caused body weight loss, irrespective of whether tolerance was induced.

Suppression of food intake over time was strongly correlated with body weight, as might be expected (Fig 6C). Food consumption is decreased in conditions associated with tolerance (DS2) or tissue injury (DS5–9; Fig EV5B), and this may therefore be part of the mechanism leading to loss of body weight in these conditions. We observed a weaker correlation between blood glucose level and weight loss (Fig EV5C and D). The causal chain here appears to be that decreased food consumption weakly correlates with decreased glucose concentration, and decreasing blood glucose is known to cause decreased body weight.

To identify candidate secreted proteins that mediate whole-organism toxin responses, we computed cumulative gene expression over 29 days for all genes that changed expression in liver and kidney respectively and calculated the Spearman correlation between their expression and the body weight on Day 29 (Fig 6D). To identify genes encoding secreted proteins, we referred to a list of 376 blood plasma proteins measured by proteomics and other methods (Nanjappa *et al*, 2014; Thul *et al*, 2017). We discovered that the four most strongly up- and downregulated genes encoding secreted proteins in the liver were all related to insulin-like growth factor-1 (Igf-1): low body weight animals consistently upregulated the Igf-1 antagonists Igfbp1 and Igfbp2 and downregulated Igf1 itself and its activator Igfals (Fig 6D; Clemmons, 2007). Three of the four genes showed similar trends in the kidney. We also found that the Spearman correlations of gene expression levels and food consumption across five time points (1, 4, 8, 15, 29 days) showed quite similar results (Fig EV5E), strongly indicating that Igf1 signaling decreased upon suppression of food consumption. These four proteins thus appear to collectively mediate strong organ-to-body communication as part of toxin responses. Igf1 promotes tissue growth (Stratikopoulos *et al*, 2008), so decreasing its activity via gene expression may cause body weight decrease in toxin-exposed rats. Igf-1 secretion also responds to blood glucose levels, so it might be acting as part of a feedback response (Clemmons, 2004, 2007). Both liver and kidney decreased Igf1 signaling, but the liver appears to

contribute to body weight loss more significantly (Figs 6E and F, and EV5F and G). This is consistent with the fact that Igf1 synthesized and secreted by the liver accounts for 75–80% of paracrine Igf1 found in the blood plasma (Clemmons, 2007).

Another endocrine factor that might play a significant role in body weight decrease is Gdf15, whose cumulative transcriptional expressions both in liver and in kidney were strongly negatively correlated with body weight (Figs 6D and EV5E). This TGFβ family member can be produced by many tissues and is known to be a toxin response gene (Zhang *et al*, 2014; Chung *et al*, 2017, p. 15; Lee *et al*, 2017). It negatively regulates feeding by binding to the receptor Gfral in the brainstem (Emmerson *et al*, 2017, p. 15; Hsu *et al*, 2017; Mullican *et al*, 2017; Yang *et al*, 2017). Animals upregulating Gdf15 transcription either in liver or in kidney decreased food consumption (Fig 6G), consistent with a role of Gdf15 in suppressing appetite. However, high expression of Gdf15 transcripts in the liver was induced only by synthetic hormones such as ethinylestradiol or tamoxifen, which induced decrease in body weight without any physiological or histopathological signs of toxicity (Fig 6G). We therefore concluded that they were causing weight loss via a mechanism that differs from classic toxin responses. When these compounds were removed, we found that Gdf15 expression in the kidney, and not the liver, responded consistently to tissue injury (Fig EV5H, Appendix Fig S7). Thus, Gdf15 expression by the kidney after tissue injury may be the general mechanism for toxin-induced weight loss, via negative regulation of feeding. Figure 7 summarizes our current view of the role of liver–body communication in toxin-induced body weight decrease, where liver-to-whole-body signals that decrease feeding remain to be identified.

## Discussion

Application of machine learning to physiology and histology data is an emerging field with high potential in translational research (Miller & Brown, 2018). We applied a fairly simple unsupervised characterization approach to discover disease states in a high-quality toxicogenomics dataset. Physiology data alone identified

**Figure 6. Low blood glucose level inhibits Igf1 paracrine to decrease body weight.**

A   Body weight change over time. Treatment conditions were grouped into the DSs on Day 29: tolerance (DS2), tissue injury (DSs5–9), other DS (DSs1,3,4), and non-DS. The number of conditions (compound x dose) in each group are shown in the brackets. Error bars are standard deviation at each time point for each group. Each normalized body weight is an average of five biological replicates.

B   Pathway activities in tolerance (71) or tissue injury associated conditions (58) compared with the rest of the conditions (222). *Y*-axis shows pathway activity scores at a given time point (signed log *P*-values from two-sample Wilcoxon test). Red and blue lines correspond to the 14 DS2-associated pathways listed in Fig EV4D and "Fer-res", respectively.

C   Relationship between cumulative food consumption over 29 days and body weight on Day 29**.**

D   Spearman correlated with cumulative gene expression in liver and kidney over 29 days and body weight on Day 29. Plasma proteome genes, four Igf1-related genes, and Gdf15 were highlighted in black, red, and blue, respectively.

E   Linear regression model with latent variables (left). Rectangle and ellipsoidal nodes indicate observed data and latent variables, respectively. Contribution of liver and kidney Igf1 transcriptional activities to body weight on Day 29, both with and without synthetic hormone data (right). Error bars are standard errors of mean calculated using lavaan R package. *P*-values of the null hypothesis that the coefficients are zero being correct were calculated by the estimates and the standard errors plugged into normal distribution. *P*-values of "l" (asterisks) are 3.1e-32 (with hormones) and 3.9e-4 (without), respectively. 97 (with hormones) and 91 (without) samples were used for the regression.

F   Cumulative Igf1 transcriptional activity over 29 days in liver and kidney, computed in the model in panel (E). Points were color-coded by the extent of body weight change. Filled squares indicate synthetic hormone treatments.

G   Gdf15 expression in liver and kidney at five time points (1, 4, 8, 15, and 29 days). Points were color-coded by changes in food consumption per day. Filled squares indicate synthetic hormone treatments.

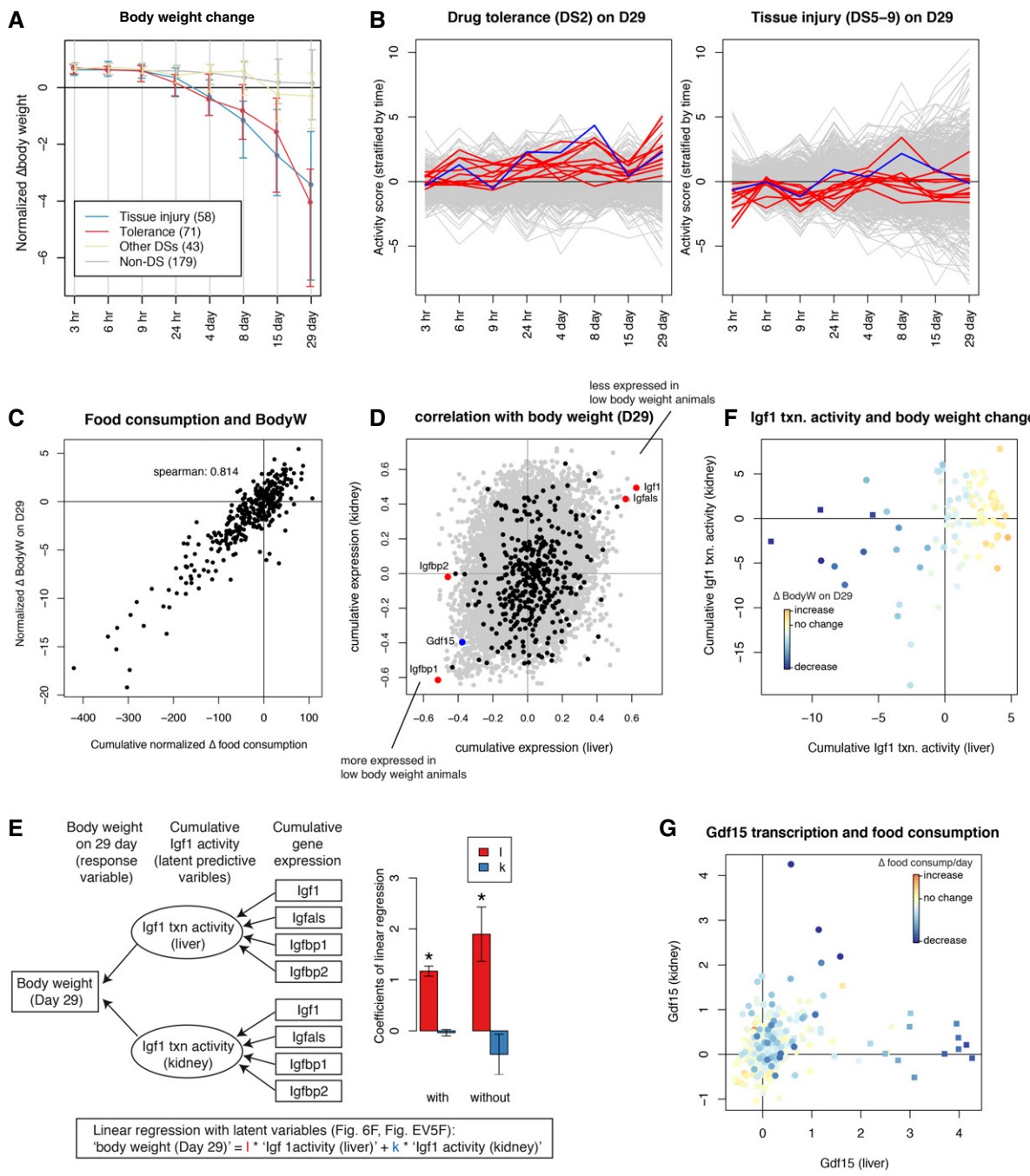

**Figure 6.**

discrete disease states (Fig 1B), and histopathology data supported these states with high statistical confidence and helped us determine their relevance to conventional toxin-induced pathologies (Figs 1C and D, and 2C). While we found some disease states that were not previously reported (i.e., DS2–4), we do not think all well-established pathological states were explained by our nine DSs at least for three reasons. First, Open TG-GATEs primarily focused on the liver and kidney pathology, so we are likely to miss phenotypes in other tissues, such as drug-induced cardiotoxicity, even if induced. Second, pathological phenotypes not induced by the 160 compounds chosen for the study should not be covered.

Third, we picked robust DSs, which were induced by ≥ 20 conditions, so any less frequent disease states were missed out in our analysis. Considering these limitations, unsupervised identification of disease states in our study is still a useful proof of concept for large-scale toxicogenomic analysis. Moreover, Open TG-GATEs includes large H&E images of most conditions, and an interesting future question is whether application of artificial intelligence machine vision approaches could help boost reliability and quantification of histopathology images.

Our approach provides a new window into the biology present in a toxicogenomic dataset, but we did not use it to try to improve

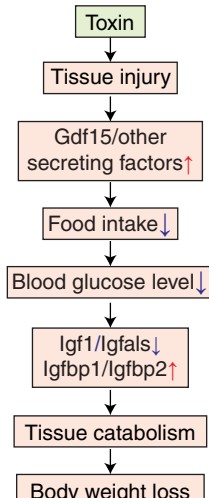

**Figure 7.  Proposed mechanism of drug-induced body weight loss.**

(1) Toxin induces tissue injury. (2) Injured tissue and the kidney secrete anorexic factors including Gdf15. (3) Hypothalamus suppresses appetite and food consumption. (4) Blood glucose concentration is decreased. (5) Production of paracrine Igf1 system in the liver is decreased. (6) Tissues, such as muscle and adipose, catabolize themselves due to suppression of glucose uptake. (7) Body weight is decreased.

predictive toxicology. That important goal might be feasible in future studies. Our immediate interest was in how well an algorithm can mimic a physician in terms of defining disease states, which is generally relevant to applications of machine learning in medicine. We were also interested in how the liver, and the organism as a whole, responds, and in some cases adapt, to continuous toxin exposure. Toxin-induced tolerance is highly relevant to environmental and pharmacologic toxicology, and to the evolution of xenobiotic defenses, but has received relatively little attention in the genomic era. Our DS analysis (Table 1) was successful in identification of standard organ pathology states used in toxicology. It also identified several non-standard states with interesting physiology, which demonstrates the potential of computational analysis in translational research and medicine.

An unexpected and interesting outcome of DS kinetics was evidence for acquisition of tolerance to xenobiotics by the liver, also known as autoprotection or pharmacokinetic tolerance. We identified DS2 as the main tolerance state. We then showed, using gene expression analysis, that tolerance is achieved by expected mechanisms that include overexpressing xenobiotic catabolizing enzymes, particularly NADPH- and GSH-dependent ones. Unexpectedly, we found that tolerance correlated with induction of biomarkers for ferroptosis resistance, using an independent dataset of cancer cells responds to drugs to identify these biomarkers. Overall, our analysis of tolerance confirms the expected role of conventional detoxification enzymes and points to ferroptosis resistance as a novel mechanism worthy of additional study. Induction of detoxification enzymes is a plausible means to protect tissues from injury by drug toxicity. For example, small molecule drug candidates have been developed that activate the toxin-responsive transcription factor NRF2 by antagonizing its interaction with KEAP (Davies *et al*, 2016). The goal of such drugs is to protect organs from damage

caused by endogenous oxidants, but they might also provide benefit in acute or chronic toxin exposure. Further research might identify targets for drugs that protect specifically against ferroptotic death. In general, induction of pharmacokinetic tolerance is an actionable next step to mitigate toxicity from exposure to environmental toxins and perhaps also endogenous toxins generated by disease processes.

Finally, we used transcriptomics to analyze possible molecular causes of body weight decrease and the role of secretory plasma proteins in orchestrating it. Despite the universal use of body weight as a biomarker of toxicity, its molecular basis is poorly understood. Body weight decrease was found in all DSs where organ injury was present, and also in the tolerance state DS2. As expected, it correlated with decreased food intake and blood glucose (Fig 6C). Remarkably, the four liver-secreted proteins that correlated best with body weight decrease were all part of the Igf1 system, and all changed in the same direction, of decreased Igf1 signaling in disease states associated with weight loss (Fig 6D). Their high correlation suggests Igf1 shutdown is a common response to diverse tissue injuries caused by toxins, where the chain of causality likely proceeded from decreased feeding causing low blood glucose levels which then cause decreased Igf1 signaling (Clemmons, 2007). Paracrine Igf1 signaling serves to increase muscle and adipose growth, so its loss may plausibly drive muscle atrophy and body weight loss (Fig 7). This leaves open the question of how organ damage triggers decreased feeding. Gdf15 is an interesting candidate. It is known as a common early marker of drug-induced liver injury (Hsiao *et al*, 2000; Chung *et al*, 2017) and binds to Gfral in the brainstem to suppress food intake (Emmerson *et al*, 2017; Hsu *et al*, 2017, p. 15; Mullican *et al*, 2017; Yang *et al*, 2017). Our data support a role of Gdf15 in mediating weight loss in response to multiple toxins (Fig 6G, Appendix Fig S7). The data also suggest that Gdf15 in response to multiple toxins is primarily synthesized in kidney, and most strongly induced by kidney toxins, consistent with a recent report that Gfral mediates weight loss in response to the kidney toxin Cisplatin (Hsu *et al*, 2017). However, Gdf15 expression by liver was only strongly correlated with weight loss in response to certain synthetic hormones that were not overt toxins (Figs 6G, and EV5G and H, Appendix Fig S7). Therefore, as yet unidentified signals are more likely to mediate liver-to-whole-body signaling to suppress food intake upon synthetic hormone treatments.

In summary, our approach shows the value of systematic data collection and analysis for revealing new organ biology. Ferroptosis emerged as an important factor in toxin responses and may be a druggable driver of tissue pathophysiology. We made progress on the little-studied problem of how organ toxicity triggers whole-body weight decrease, which is relevant to mechanistic toxicology, effects of alcohol in man, and perhaps to diagnosis and treatment of human diseases with non-toxin causes. From a translational perspective, our findings support a role of GDF15 antagonists for treatment of drug-induced cachexia. They particularly support a role of IGF1 agonists for treating cachexia more universally. Their use is complicated by glucose-lowering side effects, but we suggest they deserve more attention for treatment of cachexia across multiple diseases. Igf1 modifiers that are co-regulated during toxin-induced weight loss (Igfbp1, Igfbp2, and Igfals) are also worthy of consideration as druggable targets in cachexia.

    

# Materials and Methods

## Normalization of open TG-GATEs dataset

### Open TG-GATEs data acquisition

All but food consumption data were downloaded from the Open TG-GATEs website (https://toxico.nibiohn.go.jp/open-tggates/english/search.html) using RCurl package and parsed with XML package. Food consumption data were downloaded from another repository of Open TG-GATEs dataset at Life Science Database Archive (https://dbarchive.biosciencedbc.jp/en/open-tggates/data-11.html). An administration of one compound, one dose, and one time point is referred to as "treatment condition" or "condition" throughout the text. There were 3,564 conditions tested in total (160 chemicals, three doses, eight time points). Each condition was tested in biological quintuplicates to collect physiology (body and organ weights, blood cell counts, blood chemistry) and histology (diagnosis based on H&E staining of liver and kidney made by toxicologic pathologists) data; three of the five samples were further tested for liver and kidney microarray data; 3,528 and 975 conditions were tested for liver and kidney transcriptome.

### Drug treatment information

In the Open TG-GATEs dataset, rats were tested with one of 160 compounds (99 drugs, 55 industry toxins, six endogenous signaling molecules or metabolites), at three doses determined for each chemical (Appendix Fig S1C). After single dose treatment, animals were sacrificed at 3, 6, 9, or 24 h; after daily repeated dose treatment, animals were sacrificed on 4, 8, 15, or 29 days. Some compounds were tested in single or repeated dose treatments only, but 140 compounds were tested for all eight time points. Ninety-seven of which were at same doses, but 43 of which were tested at higher doses in single dose than at repeated dose. 365 conditions (compounds at fixed doses) were scheduled for all eight time point testing, but 14 killed animals at later time points so 351 conditions were tested at eight time points. At the time of sacrifice, physiology (hematology, body, and tissue weights) and histopathology of liver and kidney were collected for each of the five animals. Further, three of five animals representing each condition were collected and subject to liver and kidney microarray.

### Physiology data normalization

Physiology parameters (blood cell counts, blood biochemistry, and body and tissue weights) were measured for each first subject to normalization for each of 3,564 treatment conditions (compounds, doses, time points). We first averaged five biological replicates of each parameter for each condition. Because each parameter was measured in different units and they were not directly comparable, we normalized the value so that they were comparable to each other. As for normalization, we computed the mean and the interquartile range of 3,564 values of each parameter, subtracted the mean from each value, and divided by the interquartile range.

### Histopathology curation

Histopathology of liver and kidney from every treated animal was diagnosed by toxicologic pathologists. The information consists of the names of phenotype (necrosis, hypertrophy, etc.), topography (periportal, centrilobular, etc.), and grade (minimal, slight, moderate, severe). Because some histopathology phenotypes (such as necrosis) were often observed even under vehicle treatments, so they were considered independently of compound treatment. Therefore, we trimmed the histopathology observations so that we work only with observations likely induced by compound treatments. To do this, we stratified the observations into topography and grade. For each topography and grade of each phenotype, we counted the number of rats exhibits the phenotype both induced by vehicles and by compounds, and kept observations only when the ratio of the counts was more than the ratio of rats used in the project (=0.336; 5,950 and 17,685 rats used for vehicle and toxin treatments in TG-GATEs, respectively) and discarded otherwise because we do not have a firm evidence that the phenotype was induced by the toxins. After the trimming procedure, we claimed a phenotype was induced by a condition (compound, dose, time point) when at least one of the quintuplicated rats treated with the condition exhibited the phenotype.

### Transcriptome normalization

Microarray experiments were performed in three biological replicates. All the CEL files from rat liver and kidney data were downloaded from the Open TG-GATEs website. There were 14,143 and 3,905 CEL files for the liver and kidney. The CEL files of the same tissue were handled simultaneously for computing a normalized expression matrix using affy, affyio, BufferedMatrix, BufferedMatrixMethods, rat2302.db packages. Normalization was performed by robust multiarray analysis of BufferedMatrix.justRMA () function of BufferedMatrixMethods, which log2-transformed the resulting expression profiles. Three biological replicates were averaged to produce an expression profile for each condition, and a profile of the corresponding vehicle treatment was subtracted. This gives expression profiles of 3,528 and 975 conditions in liver and kidney, respectively.

### Food consumption data normalization

For 337 conditions (132 unique compounds), food consumption was measured at nine time points (1, 4, 8, 11, 15, 18, 22, 25, and 29 days). For these conditions, food consumption of rats administered with compounds was subtracted from that of rats administered with vehicles.

## Identification of disease states/physiology and histology overrepresentation

### Computing physiology t-SNE

Using 1-Pearson correlation (also known as "Pearson distance") as distance measure between any pairs of treatment conditions in the physiology space, we first computed a distance matrix across 3,564 conditions. We next set a seed $i$ for random number generator (RNG) ($i = 1$–100) and ran t-SNE based on the calculated distance matrix using Rtsne() function in Rtsne package, to generate a 2-dimensional coordinate of each conditions on the t-SNE map.

### Filtering disease-associated conditions

Severity scores were computed by counting co-occurring histology phenotypes for liver and kidney and mapped onto t-SNE map. Two-dimensional density landscape of severity scores was computed

  

using bkde2D() function in KernSmooth package. Severity score is recomputed by estimating the severity score from the 2-dimensional density map using interp.surface() function in fields package. Conditions containing higher severity scores than an arbitrary threshold were considered to be associated with some diseases and further selected for disease identification.

### Clustering for identifying disease states

Conditions with higher severity scores were clustered based on their t-SNE coordinates using density-based clustering of applications with noise (DBSCAN). This is achieved by dbscan() function in dbscan package. 100 runs from t-SNE to clustering with different RNG seeds were summarized by ensemble clustering using cl_consensus() function in clue package. This identified 15 clusters that contain 5–203 conditions. To gain robust disease states that are induced by multiple compounds, we discarded smaller clusters composed of fewer than 20 conditions or induced only by one compound, because we expected that such small clusters do not have strong statistical power due to the small sample size in further transcriptome analysis. We recomputed the memberships and likelihoods to limit our interest to larger clusters with $\geq 20$ conditions and found nine consensus clusters in total ranging from 37 to 203 conditions (10–55 unique compounds). At the same time, 2,723/3,564 conditions were identified a non-disease states.

## Characterization of physiology and histology of nine DSs

### Relative severity between liver and kidney

Liver and kidney severity scores for each disease were compared to assess which tissue was more affected in terms of histopathology. Relatively affected tissue was assessed by scatter plot (Fig 2A, top) as well as log ratio: $\log_{10}(severity_{liver}) - \log_{10}(severity_{kidney})$ (Fig 2A, bottom).

### Deviation of physiological parameters in each DS

Changes in physiology parameters were assessed by unpaired two-sample two-sided Wilcoxon test between conditions in each DS and conditions in non-DS. Resulting $P$-values were adjusted to false discovery rate (FDR; also known as $q$-values) and further converted to "signed log $q$-values" (Shimada *et al*, 2016; Fig 2B). Physiological parameters whose $q$-value $< 10^{-10}$ against at least one DS were shown in Fig 2B.

### Relative enrichment of histopathological phenotypes among DSs

Among conditions associated with at least one histopathological observation, we assessed whether each histopathology phenotype was more observed in a specific DS, using one-sided Fisher's exact test. All the $P$-values were FDR-adjusted and converted to singed log $q$-values, and histopathology phenotypes whose $q$-values $< 5 \times 10^{-3}$ against at least one DS are shown in Fig 2C.

## Elastic net classification of DS using microarray data

To assess whether liver or kidney transcriptome is powerful enough to distinguish each DS from the rest, we built elastic net classifiers using cv.glmnet() function of glmnet package. The performance of an elastic net classifier built for each tissue and each DS was tested as follows: For each DS, conditions (whose transcriptome was available) were either assigned into the DS or not. Those assigned and those not were, respectively, split into 10 bins of the same sizes randomly (i.e., 10 groups for the DS, 10 groups for not). An elastic net classifier was then trained with one of the 10 groups being left out for both, where the conditions were weighted reciprocally proportional to the two sizes (# of the DS or not). Binomial family for the response type and area under curve for the type measure were used for elastic net. The left-out conditions were used as testing data for the trained classifier. This 10-fold cross-validation was repeated 10 times, with different ways to split conditions into 10 bins, and the prediction probabilities across the 10 runs were averaged (repeated cross-validation, also known as prevalidation). Performance of the classification was assessed by area under receiver operator curve (AUROC) computed using auc() function of pROC package. Finally, to evaluate the significance of the classification based on the identified disease states, the same procedure was run on randomly withdrawn conditions with the same sample sizes and AUROC values were compared. AUROC values based on identified DSs were substantially higher than those based on randomly withdrawn conditions.

## Pathway analysis

### Compiling pathways

We assembled 973 pathway information using KEGG.db (v3.2.3) and GO.db (v3.5.0) Bioconductor packages. Using rat2302.db (v3.2.3) and org.Rn.eg.db (v3.5.0) packages, we found 914 of which have $\geq 10$ genes that were measured in Affymetrix Rat Genome 230 2.0 Array.

### Computing activity scores

We assessed whether the 914 GO and KEGG pathways were activated or inactivated across conditions assigned to each DS, compared to non-DS conditions. We assumed that some pathways changed exclusively in one DS, while other pathways changed in multiple DSs. To appropriately capture this, we modified gene set enrichment analysis (GSEA; Subramanian *et al*, 2005). GSEA sorts entire genes based on their expression and performs one-sample Kolmogorov–Smirnov (KS) test, a permutation-based test to assess the significance of KS statistics. In our method, we first computed a KS statistic (also known as "enrichment score" in GSEA) of each pathway in every condition and asked whether the enrichment score of conditions assigned to one DS is overall higher or lower compared to non-DS conditions, using two-sample Wilcoxon test (also known as Mann–Whitney $U$-test). Resulting $P$-values were converted to signed $\log_{10}$ $P$-values, which we termed "activity scores". A large positive or negative activity score indicates that a pathway is significantly up- or downregulated across conditions assigned in the DS compared to non-DS conditions. Note that we decided to not adjust $P$-values for multiple hypothesis testing for transcriptome analysis because pathway information from two different databases, GO and KEGG, is highly redundant, but instead we chose fairly strict criteria ($P \leq 1 \times 10^{-5}$) for calling a pathway's change significant.

## Transcriptome characterization of DSs

### DSs similarity based on transcriptome

Using 723 liver and 192 kidney pathways whose transcriptional activity was significantly changed at least in one DS, we measured

similarity of the transcriptome of the nine DSs using hierarchical clustering, using 1-Spearman correlation as distance measure and a complete linkage method for the clustering (Fig 3B).

*Transcriptional characterization of DSs*

We mapped 723 pathways in the liver transcriptome based on their activities across six DSs (DS1–2,5–8) whose liver transcriptome was substantially deviated from non-DS. We first checked if each pathway's expression changed in the liver of the six DSs, by comparing their activity score with thresholds ($\geq 5$ for upregulation, $\leq -5$ for downregulation). Then, patterns of up/downregulations of a pathway were matched with the dendrogram (Appendix Fig S5C–E). Pathways exclusively changed in one direction only in one DS were mapped onto each DS (e.g., xenobiotic metabolism in DS2), and pathways commonly changed in multiple DSs were associated with the corresponding branching point in the dendrogram (e.g., cancer signature in DS6–7). In the extreme, a pathway upregulated in all the six DSs ["large ribosomal subunit" (GO:0042273)] was associated with the top branching point in the dendrogram. Note that there were some pathways that were significantly changed in one or more pathways but not mapped to the dendrogram. For example, "Terpenoid biosynthesis (rno00900)" was upregulated in three DSs (DS1,2,8), but there was no equivalent point in the dendrogram.

### Disease transition network between DSs

Of the 365 conditions (compounds and doses) scheduled at all eight time points between 3 h to 29 day (14 of which were scheduled at eight time points but rats were killed by compounds before 15-day or 29-day time points, so 351 of 365 were actually tested at the eight time points; we included these dead rats in this analysis), we looked at the DS assigned at each time point. 119 conditions did not exhibit any DSs. Of the 246 conditions that took some DSs at least once, 90 took more than one DS across eight time points. In some cases, non-DS states were observed while transitioning from one DS to another. In the dynamics between DSs, however, we visualized them as directly transitioning from one DS to another, to highlight the relationship between DSs. The dynamics between DS was visualized using igraph package (Fig 4B). Transition to and from non-DS (represented by outer open circle) were manually added in Adobe Illustrator.

### Enzyme stratification by cofactors

The 14 pathways upregulated exclusively in DS2 contain various xenobiotic metabolism enzymes encoding genes. We classified these genes based on their Enzyme Commission (EC) numbers, which were available in org.Rn.eg.db package. All of these enzyme-encoding genes were oxidoreductases (EC1), transferases (EC2), or hydrolases (EC3), which require cofactors for the enzymatic functions. Except for EC3, which requires water as cofactor that is abundant in cells, we regrouped EC1 and EC2 enzymes based on the cofactors: NAD(P)H (EC1.1.1.1, EC1.1.1.10, EC1.1.1.14, EC1.1.1.21, EC1.1.1.22, EC1.1.1.30, EC1.1.1.42, EC1.1.1.44, EC1.1.1.45, EC1.1.1.49, EC1.1.1.62, EC1.1.1.63, EC1.1.1.64, EC1.1.1.105, EC1.1.1.146, EC1.1.1.149, EC1.1.1.205, EC1.1.1.270, EC1.1.1.284, EC1.2.1.3, EC1.2.1.5, EC1.2.1.8, EC1.2.1.31, EC1.2.1.36, EC1.2.1.47, EC1.3.1.2, EC1.3.1.3, EC1.3.1.24, EC1.5.1.30, EC1.8.1.7, EC1.8.1.9, EC1.11.1.6, EC1.11.1.9, EC1.11.1.12, EC1.14.13.8, EC1.14.13.17, EC1.14.13.100,

EC1.15.1.1, EC1.17.1.4), cytochrome (EC1.10.2.2), oxygen (EC1.1.3.8, EC1.2.3.1, EC1.3.3.3, EC1.3.3.4, EC1.4.3.4, EC1.16.3.1, EC1.17.3.2), disulfide (EC1.8.4.2, EC1.17.4.1), flavin (EC1.14.14.1), iron-sulfur (EC1.14.15.3, EC1.14.15.4, EC1.14.15.5, EC1.14.15.6), S-adenosyl methionine (EC2.1.1.6, EC2.1.1.67), acyl-CoA (EC2.3.1.5, EC2.3.1.15, EC2.3.1.20, EC2.3.1.37, EC2.3.1.75, EC2.3.1.76, EC2.3.1.135, EC2.3.2.2, EC2.3.2.4), nucleotide sugar (EC2.4.1.17, EC2.4.1.22, EC2.4.1.38, EC2.4.1.50, EC2.4.1.66, EC2.4.1.90, EC2.4.1.109, EC2.4.1.152, EC2.4.1.221, EC2.4.1.222, EC2.4.2.3, EC2.4.2.8, EC2.4.2.10, EC2.4.99.1, EC2.4.99.6), glutathione (EC2.5.1.18, EC2.5.1.61), ATP (EC2.7.1.17, EC2.7.1.21, EC2.7.1.48, EC2.7.4.9, EC2.7.7.9, EC2.7.11.22). Expression of the genes stratified by cofactors in DS2 was assessed by two-sample Wilcoxon test (Fig 5A). Furthermore, GSEA was performed to see enrichment of NAD(P)H-dependent, GSH-dependent, and all enzyme-encoding gene expressions against DS2 transcriptome.

### Discovery of biomarkers to different cell death phenotypes

We previously showed that cell-line selectivity of lethal compounds (i.e., growth inhibitory ($GI_{50}$) profiles across cell lines) in the NCI-60 dataset can explain their lethal mechanism of action and that 2,565 cell-line selective lethal compounds were clustered into 18 mechanistically distinct classes (Shimada *et al*, 2016). While most of their mechanisms of action were not fully characterized, yet a few annotated ones were DNA-targeting compounds (DNA), ferroptosis (Fer), and tyrosine kinase inhibitors (TKI). In the paper, we also correlated basal microarray expression profiles with drug sensitivity profiles of each mechanism class, where positive and negative correlations can be interpreted as more abundant in resistant or sensitive cell lines, respectively. We extended this approach in this study. We took top 200 most positively and negatively correlated genes in the three cell death phenotypes (DNA, Fer, TKI) as resistant (res) and sensitive (sen) biomarkers for the phenotype, because these are genes likely overexpressed in cells resistant or sensitive to each cell death phenotype. Thus, we created six human gene sets: DNA-res, DNA-sen, Fer-res, Fer-sen, TKI-res, and TKI-sen. We then converted the genes to orthologous genes in rats using Ensembl's biomart and Mouse Genome Informatics (http://www.informatics.jax.org/downloads/reports/HOM_AllOrganism.rpt) and found 158–185 rat orthologs for each gene set. We computed activity scores of these six gene sets across nine disease states to assess whether they are resistant or sensitive to the three different mechanisms of cell death (Fig 5C).

### Correlation of pathways with ferroptosis sensitivity signatures

We computed enrichment scores of the two curated gene sets, Fer-sen and Fer-res, and computed Spearman correlation of enrichment scores between Fer-sen/Fer-res and the 914 GO and KEGG pathways. We found most of the 14 pathways exclusively upregulated in DS2 were the most highly correlated with Fer-res, while they were somewhat negatively correlated with Fer-sen (Fig EV4C and D).

### Pathway activity analysis in time course

Of the 351 conditions (compounds and doses) that were tested at all eight time points, 71 were assigned to DS2 and 58 to DS5–9 on 29-day time point (Fig EV5A). They were named as "tolerance" and

"tissue injury", respectively. The other 222 conditions were either other DSs or non-DS. At each time point, we assessed whether pathways were changed in tolerance or tissue injury. We computed of each pathway per condition and assessed whether the statistics were deviated between the classes (tolerance vs. others; tissue injury vs. others) using two-sample two-sided Wilcoxon test (Fig 6B).

**Compiling blood plasma proteins**

We assembled the collection of experimentally validated plasma proteins from two different databases. First, from Human Protein Atlas (https://www.proteinatlas.org/), 3,704 "predicted secretory proteins" have evidences at protein levels were looked at, 2,960 of which have rat orthologs. Second, from Plasma Proteome Database (http://plasmaproteomedatabase.org/), 468 proteins have more than one associated reference that they were observed in plasma, 382 of which have rat orthologs. Altogether, 376 rat orthologs (376 unique Entrez IDs) were observed from the two databases that were also measured in Affymetrix Rat Genome 230 2.0 Array.

**Correlation between gene expression and food consumption or body weight**

We computed area under curve (weighted sum) of the all genes' expression in liver and kidney over 29 days, including 376 plasma protein-encoding genes for 351 conditions tested at all time points. Then, we calculated the Spearman correlation coefficients between the cumulative gene expression and the body weight on 29 day (Fig 6D). The positive and negative correlation indicates genes were expressed less or more in the animals with decreased body weight. We also computed Spearman correlation coefficients between individual gene expression, not cumulative, and food consumption per day (Dataset EV9). In which, measurements at five time points (1, 4, 8, 15, 29 days) were treated as independent conditions, and correlation was calculated across all conditions and time points (Fig EV5E).

**Linear regression of body weight or food consumption on Gdf15 or Igf1 activities**

*Igf1 and body weight*
To assess the relationship between Igf1 and body weight, transcriptional activity of Igf1 system (named as "Igf1 transcriptional activity" in Fig 6E) in liver and kidney was summarized from four Igf1-related genes in the tissues as latent variables, which were further used to regress changes in body weight on 29-day time point (Fig 6E). This latent variable analysis was performed using lavaan package. In the model, "Igf1 activity" was conceived as a latent variable each for liver and kidney, which is estimated from cumulative expression of four Igf1-related genes (Igf1, Igfals, Igfbp1, and Igfbp2). And the two latent variables, Igf1 activity for liver and kidney, were used to see their contributions in the change in body weight on 29 day.

*Gdf15 and food consumption*
Multivariate linear regression of food consumption on Gdf15 expression in liver and kidney was performed. Since Gdf15 level in the tissues are substantially different among DSs, the linear regression

was also performed with the data stratified into five DSs [tolerance (DS2), liver injury (DS5–7), kidney injury (DS9), bleeding (DS8), non-injury(DS1,3,4)], where DSs were taken as a categorical interaction term in lm() (Appendix Fig S7). The significance of the coefficients stratified by DSs was plotted in Fig EV5H.

Processed data from Open TG-GATEs are available in the following datasets:

- All drug treatment conditions: Dataset EV1
- Normalized physiology data: Dataset EV2
- Liver and kidney histopathology data: Dataset EV3
- Disease states and corresponding treatment conditions: Dataset EV4
- Liver normalized transcriptome data: Dataset EV5
- Kidney normalized transcriptome data: Dataset EV6
- Significantly up- and downregulated pathways in disease states of rat livers in Open TG-GATEs: Dataset EV7
- List of xenobiotic metabolism genes whose expression was upregulated in disease state DS2: Dataset EV8
- Food consumption data from Open TG-GATEs: Dataset EV9

**Expanded View** for this article is available online.

## Acknowledgements

We thank Rebecca Ward, Laura Maliszewski, Debora Marks, Peter Sorger, and Peter Koch of Harvard Medical School; Scott Dixon of Stanford University; Akiyoshi Suganuma, Kappei Tsukahara, Etsuko Ohta, and Kenji Kubara of Eisai Tsukuba Research Laboratories; and Dahlene Fusco of Massachusetts General Hospital for helpful discussion. This study was supported by JSPS Overseas Research Fellowships (to K.S.) and a NIH grant 5P50GM107618.

## Author contributions

KS: Conceptualization, formal analysis, investigation, data curation, writing, visualization, and project administration. TJM: conceptualization, writing, and supervision.

## Conflict of interest

The authors declare that they have no conflict of interest.

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
