## [Review Process File · Molecular Systems Biology]

Unsupervised identification of disease states from high dimensional physiological and histopathological profiles

Kenichi Shimada and Timothy J. Mitchison.

Review timeline:	Submission date:	2 nd September 2018
	Editorial Decision:	26 th October 2018
	Revision received:	25 th November 2018
	Editorial Decision:	20 th December 2018
	Revision received:	14 th January 2019
	Accepted:	21 st January 2019

Editor: Thomas Lemberger

Transaction Report:

1st Editorial Decision

26th October 2018

Dear Dr Shimada,

Thank you again for submitting your work to Molecular Systems Biology. We have now heard back from the three referees who accepted to evaluate the study. As you will see, the referees find the topic of your study of potential interest and are supportive. They raise however a series of points, which should be convincingly addressed in a revision of the work. The indications provided by the reviewers are very clear in this regard.

We would kindly ask you to also consider the following more editorial points:

- Please remove any jargon from the text, title and abstract. Given that t-SNE has been published 10 years ago and its use is currently widespread, reference to 'advanced machine learning' as well as the emphasis on 'machine characterization' seem difficult to justify, and we would kindly ask you to modify the text and title accordingly.

REFeree REPORTS

Reviewer #1:

The authors examined xenobiotic-induced tissue injury mechanisms. They used unsupervised machine learning to classify toxin-induced effects into nine states. They found that these states reveal that tolerance develops over time in many cases. Specifically, they found that ferroptosis biomarkers decline over time, and that this correlates with tolerance. They also analyzed cachexia induced by organ injury and found that GDF15 and IGF1 mediate this injury-induced cachexia. Interestingly, they found that certain drug classes mapped onto specific disease states. Overall, this is an exciting study that provides a new model for xenobiotic-induced tissue injury and tolerance, and links ferroptosis to tolerance. This will be of significant interest to many researchers who study drug toxicity and drug mechanisms in general. My only suggestion is that it would be useful for the

authors to discuss more directly the actionable steps based on their findings, in terms of mitigating drug toxicity.

Reviewer #2:

Shimada & Mitchison present an interesting re-analysis of the Open TG-GATEs database, focused on the use of physiological data to first define unique toxin-induced 'disease states', followed by various analyses of those states using histopathological and gene expression data. The authors discover several "disease states" (note: the authors mention that these don't always correspond to actual diseases, but do not find this too confusing) that have not previously been documented in the literature, and make connection between toxin adaptation and 'ferroptosis', as well as body weight decrease and the (putatively) secretion of various growth factors. Most of the data is analyzed at a high level of abstraction (correlations between normalized data and clustered data/GO terms), so it is difficult to be certain in many cases of the actual strength of the biological phenotype in the animal, but the authors appear to have made some interesting connections and this work should be considered for publication with appropriate modifications.

Major:

-At various points throughout the manuscript the authors make rather strong claims that do not appear to be backed up by specific data points or published studies, or at least not in the text as currently written. For example, on line 149/150 it is stated that "DS1-4 were not associated with detectable tissue injury, and would likely elude a conventional toxicogenomic analysis". It is not clear on what basis this statement can be made in the Results. Likewise, on Line 168 it is stated that "DS7 corresponds to hepatocellular injury", and in Line 172 "DS9 corresponds to kidney injury". By what specific criteria?

-The authors state that they recovered known disease state categories, and also discovered new ones that were not previously reported (e.g. DS1-4). However, they have not addressed whether there are well-established pathological states that were not recovered by their analysis. I.e., this analysis found some things that were known before, but did this analysis also miss anything that was known? If so, why?

-The authors repeatedly invoke the use of 'machine characterization' (title), 'machine learning' (abstract, Results), and 'machine-mimic' and 'machine-identified' (Introduction). Are these terms synonymous? In the case of machine learning, it is not clear that this term is being used correctly. For example, have the authors employed a subset of the entire dataset to generate a model classifier using ML methods, then applied that model to a held-out (or distinct) dataset to see whether the model makes accurate prediction on new data? The addition of a true test of whether these models enabled improved classification and prediction of disease states using an independent dataset would be a great addition to the paper, if such an analysis was feasible.

For the rest, it may be technically true that the various analyses were performed using a machine, but this is also true of many analyses, both simple and complex, and the terms employed (e.g. machine-identified, machine-mimic) do not seem to add anything except a distracting buzz-word. Indeed, as the authors write in the Discussion (line 381-2): "we applied a fairly simple unsupervised characterization approach..."

-In Line 340-350 and 360-370 of the Results (and associated Figure 6) the authors repeatedly describe changes in transcript levels as changes in protein levels. They have only analyzed relative gene expression, not protein secretion, and should take more care in describing and properly annotating gene versus protein names (e.g. gene names are italicized) and what they have truly observed. All speculations about what changes in gene expression might mean for protein secretion should be saved for the Discussion.

-The correlation between Gdf15 mRNA expression and weight appears is reported to be quite low (-0.2). Thus, it seems unlikely to "be the general mechanism for toxin-induced weight loss". At the very least, such statement based on correlations to gene expression, without actual interventional studies, should be tempered in the Results section and potentially also the Discussion.

Minor:

- The entire work could use another round of editorial correction. There are numerous places where a word is missing, or an extra word is interested. E.g. Line 36, missing 'the' after 'limits and'.
- Lines 122 - 127 are extremely confusing due to multiple missing or extra words and need to be re-written.
- Line 130/131. Is 'cluster calling 100 times...' what is meant?

Reviewer #1:

The authors examined xenobiotic-induced tissue injury mechanisms. They used unsupervised machine learning to classify toxin-induced effects into nine states. They found that these states reveal that tolerance develops over time in many cases. Specifically, they found that ferroptosis biomarkers decline over time, and that this correlates with tolerance. They also analyzed cachexia induced by organ injury and found that GDF15 and IGF1 mediate this injury-induced cachexia. Interestingly, they found that certain drug classes mapped onto specific disease states. Overall, this is an exciting study that provides a new model for xenobiotic-induced tissue injury and tolerance, and links ferroptosis to tolerance. This will be of significant interest to many researchers who study drug toxicity and drug mechanisms in general. My only suggestion is that it would be useful for the authors to discuss more directly the actionable steps based on their findings, in terms of mitigating drug toxicity.

We appreciate this positive comment of Reviewer #1. Her/his suggestion was to discuss more directly the actionable steps based on their findings, in terms of mitigating drug toxicity. Accordingly, we

added the following paragraph in the discussion (Lines 432–441):

Induction of detoxification enzymes is a plausible means to protect tissues from injury by drug toxicity. For example, small molecule drug candidates have been developed that activate the toxin-responsive transcription factor NRF2 by antagonizing its interaction with KEAP (Davies et al (2016) J Med Chem 59:3991). The goal of such drugs is to protect organs from damage caused by endogenous oxidants, but they might also provide benefit in acute or chronic toxin exposure. Further research might identify targets for drugs that protect specifically against ferroptotic death. In general, induction of pharmacokinetic tolerance is an actionable next step to mitigate toxicity from exposure to environmental toxins, and perhaps also endogenous toxins generated by disease processes.

Reviewer #2:

Shimada & Mitchison present an interesting re-analysis of the Open TG-GATEs database, focused on the use of physiological data to first define unique toxin-induced 'disease states', followed by various analyses of those states using histopathological and gene expression data. The authors discover several "disease states" (note: the authors mention that these don't always correspond to actual diseases, but do not find this too confusing) that have not previously been documented in the literature, and make connection between toxin adaptation and 'ferroptosis', as well as body weight decrease and the (putatively) secretion of various growth factors. Most of the data is analyzed at a high level of abstraction (correlations between normalized data and clustered data/GO terms), so it is difficult to be certain in many cases of the actual strength of the biological phenotype in the animal, but the authors appear to have made some interesting connections and this work should be considered for publication with appropriate modifications.

We thank Reviewer #2 for her/his deep understanding of our work. We do agree with her/him on that our correlational study uncovers general trends within the large dataset while it does not necessarily lead to the understanding of precise mechanisms of action of individual compounds.

Major:

-At various points throughout the manuscript the authors make rather strong claims that do not appear to be backed up by specific data points or published studies, or at least not in the text as currently written. For example, on line 149/150 it is stated that "DS1-4 were not associated with detectable tissue injury, and would likely elude a conventional toxicogenomic analysis". It is not clear on what basis this statement can be made in the Results. Likewise, on Line 168 it is stated that "DS7

corresponds to hepatocellular injury", and in Line 172 "DS9 corresponds to kidney injury". By what specific criteria?

We are sorry for the confusion. In the text, we tried to give a brief description of the nine DS accompanying Table 1, however, the sentences pointed out by the reviewer were too simple and lost clarity. The correspondence of each DS to known clinical phenotypes was according to physiological and histopathological characterization of each DS, as in standard clinical practice. So we replaced the sentences to the following:

Regarding DS1-4: “While DS1-4 induced systemic physiological and pathological changes, they were not associated with detectable tissue injury or exhibited transcriptional response to proinflammatory cytokines, and would likely elude a conventional toxicogenomic analysis.”

Regarding DS6-7 and DS9: DS6 exhibited DBIL increase and various periportal histopathology phenotypes, which corresponds to cholestasis, an injury of the liver bile ducts. DS7, on the other hand, exhibited hepatocellular injury such as increase in blood AST, ALT, and LDH levels as well as single cell necrosis in the liver. ... DS9 is marked by BUN increase and hypertrophy and neutrophil infiltration in kidney, indicating kidney injury. The database contained fewer reference kidney toxins than liver toxins, perhaps explaining why we observed only one disease state that mapped to kidney pathology (Schrier *et al*, 2004) in the analysis.

-The authors state that they recovered known disease state categories, and also discovered new ones that were not previously reported (e.g. DS1-4). However, they have not addressed whether there are well-established pathological states that were not recovered by their analysis. I.e., this analysis found some things that were known before, but did this analysis also miss anything that was known? If so, why?

We thank the reviewer for this elaborate question. Yes, it is possible that we may miss important disease states for at least three reasons. First, Open TG-GATEs primarily focused on the liver and kidney pathology, so we are likely to miss phenotypes in other tissues, such as drug-induced cardiotoxicity, even if induced. Second, pathological phenotypes not induced by the 160 compounds chosen for the study should not be covered. Third, we picked robust DSs, which were induced by ≥ 20 conditions (condition = compound x dose x time), so any less frequent disease states were missed out in our analysis. We made this point explicit in the text.

-The authors repeatedly invoke the use of 'machine characterization' (title), 'machine learning' (abstract, Results), and 'machine-mimic' and 'machine-identified' (Introduction). Are these terms synonymous? In the case of machine learning, it is not clear that this term is being used correctly. For example, have the authors employed a subset of the entire dataset to generate a model classifier using ML methods, then applied that model to a held-out (or distinct) dataset to see whether the model makes accurate prediction on new data? The addition of a true test of whether these models enabled improved classification and prediction of disease states using an independent dataset would be a great addition to the paper, if such an analysis was feasible. For the rest, it may be technically true that the various analyses were performed using a machine, but this is also true of many analyses, both simple and complex, and the terms employed (e.g. machine-identified, machine-mimic) do not seem to add anything except a distracting buzz-word. Indeed, as the authors write in the Discussion (line 381-2): "we applied a fairly simple unsupervised characterization approach..."

We thank both the editor and this reviewer for pointing this out. These terms were meant to be synonymous. In the revised text we changed all of them to “machine learning”. In our study, supervised classification was performed for only one piece of data (Appendix fig. S7A), in which performance of an elastic net classifier was examined using 10-fold cross validation. The rest of analysis used an unsupervised method, and the method we chose, or t-SNE, is a nonlinear dimensionality reduction technique in which the coordinates in low dimension space are determined by the data points taken into consideration, but not designed to project new data points onto the same low dimension space, so extrapolating new dataset was not possible. While we could build another classifier based on our disease states and apply to a held-out or distinct dataset was possible, that we believe is out of the scope of this study. Our intention was rather to see this study as a correlational one and we tried to integrate multiple different kinds of data (physiology, histopathology, transcriptome), and try to find the partitions that represents different disease states that make sense. We do think such unsupervised clustering is considered a domain of machine learning.

-In Line 340-350 and 360-370 of the Results (and associated Figure 6) the authors repeatedly describe changes in transcript levels as changes in protein levels. They have only analyzed relative gene expression, not protein secretion, and should take more care in describing and properly annotating gene versus protein names (e.g. gene names are italicized) and what they have truly observed. All speculations about what changes in gene expression might mean for protein secretion should be saved for the Discussion.

We thank this reviewer for pointing this out. We agree that the indicated sentences are confusing. We modified the text to clarify that all the changes observed are transcriptional.

-The correlation between Gdf15 mRNA expression and weight appears is reported to be quite low (-0.2). Thus, it seems unlikely to "be the general mechanism for toxin-induced weight loss". At the very least, such statement based on correlations to gene expression, without actual interventional studies, should be tempered in the Results section and potentially also the Discussion.

We agree with this reviewer that (statistically significant but) weak correlation between Gdf15 mRNA expression and food consumption may not sound convincing to explain the role of Gdf15 in food consumption. However, since Gdf15 expressed in any tissues can contribute to increase in total blood Gdf15 protein level and to induce its effect, we think the correlation between Gdf15 mRNA expression in any single tissue and food consumption may not be significant. Therefore, we moved most of the figures related to Gdf15 to supplemental (extended view and appendices), except for the relationship between liver and kidney Gdf15 expression and food consumption, which we would like to highlight. Gdf15 itself, and antagonist antibodies, are under development for treatment of various diseases, so there is currently great interest in any physiology data on this interesting signaling factor.

Minor:

-The entire work could use another round of editorial correction. There are numerous places where a word is missing, or an extra word is interested. E.g. Line 36, missing 'the' after 'limits and'.

We ask for editorial correction if necessary.

-Lines 122 - 127 are extremely confusing due to multiple missing or extra words and need to be re-written.

We changed description accordingly.

-Line 130/131. Is 'cluster calling 100 times...' what is meant?

We are sorry for the confusion. We changed the text accordingly to following:

While t-SNE has been widely used due to its power to highlight the heterogeneity of the high-dimensional data in a lower dimensional space, its stochastic algorithm gives an output that are

similar but slightly distinct from each other, depending on the pseudorandom number generator it uses. To compensate this stochastic nature of t-SNE, we ran t-SNE and clustering with different pseudorandom generator seeds iteratively for 100 times and sought for “consensus clusters” across the 100 different sets of clusters (*Hornik, 2005*).

2nd Editorial Decision

20th December 2018

Thank you again for submitting your revised work to Molecular Systems Biology. We are now globally satisfied with the modifications made and I am pleased to inform you we will be able to accept your paper for publication pending the minor modifications below:

- we would suggest to change the title of your study to highlight better the approach used: "Unsupervised identification of disease states from high dimensional physiological and histopathological profiles". Would this work? We would prefer to de-emphasize the role of machine learning since the approach is based on clustering (even if clustering is a form of machine learning). The mention of the identification of mechanisms of resistance in the title might also be too strong a claim given that the hypotheses generated would still need to be followed up to be verified.

2nd Revision - authors' response

14th January 2019

Thank you very much for accepting our manuscript (Manuscript #: MSB-18-8636) for a publication on *Molecular Systems Biology* upon minor modifications.

3rd Editorial Decision

21st January 2019

Thank you again for sending us your revised manuscript. We are now satisfied with the modifications made and I am pleased to inform you that your paper has been accepted for publication.

Corresponding Author Name: Kenichi Shimada
Manuscript Number: MSB-18-8636